# Interruption of an *MSH4* homolog blocks meiosis in metaphase I and eliminates spore formation in *Pleurotus ostreatus*

Brian Lavrijssen[1], Johan P. Baars[1], Luis G. Lugones[2], Karin Scholtmeijer[1]*,
Narges Sedaghat Telgerd[1], Anton S. M. Sonnenberg[1], Arend F. van Peer[1]

1 Plant Breeding Department, Wageningen University and Research, Wageningen, The Netherlands,
2 Microbiology Faculty of Science, Utrecht University, Utrecht, The Netherlands

* karin.scholtmeijer@wur.nl

**Data Availability Statement:** All relevant data are within the manuscript and its Supporting Information files.

## Abstract

*Pleurotus ostreatus*, one of the most widely cultivated edible mushrooms, produces high numbers of spores causing severe respiratory health problems for people, clogging of filters and spoilage of produce. A non-sporulating commercial variety (SPOPPO) has been successfully introduced into the market in 2006. This variety was generated by introgression breeding of a natural mutation into a commercial variety. Our cytological studies revealed that meiosis in the natural and derived sporeless strains was blocked in metaphase I, apparently resulting in a loss of spore formation. The gene(s) underlying this phenotype were mapped to an 80 kb region strongly linked to sporelessness and identified by transformation of wild type genes of this region into a sporeless strain. Sporulation was restored by re-introduction of the DNA sequence encoding the *P. ostreatus* meiotic recombination gene *MSH4* homolog (*poMSH4*). Subsequent molecular analysis showed that *poMSH4* in the sporeless *P. ostreatus* was interrupted by a DNA fragment containing a region encoding a CxC5/CxC6 cysteine cluster associated with Copia-type retrotransposons. The block of meiosis in metaphase I by a *po*MSH4 null mutant suggests that this protein plays an essential role in both Class I and II crossovers in mushrooms, similar to animals (mice), but unlike in plants. *MSH4* was previously shown to be a target for breeding of sporeless varieties in *P. pulmonarius*, and the null mutant of the *MSH4* homolog of *S. commune (scMSH4)* confers an extremely low level of spore formation. We propose that *MSH4* homologs are likely to be a breeding target for sporeless strains both within *Pleurotus* sp. and in other *Agaricales*.

## Introduction

During the cultivation of mushrooms, fruiting bodies can release a large number of spores. These spores can cause severe problems for people harvesting and handling the crop. Repeated exposure to high spore numbers causes extrinsic allergic alveolitis, an inflammation of the alveoli in the lung provoked by inhalation of spores [1]. In addition, spores clog filters in the climate-control system and play an important role in spreading viral diseases [2, 3]. *Pleurotus ostreatus* (Oyster mushroom), one of the most widely cultivated edible mushrooms [4], is

**Funding:** The authors received no specific funding for this work.

**Competing interests:** The authors have declared that no competing interests exist.

especially known for its heavy sporulation, and spore densities of $10^{10}$ spores/m$^3$ are easily achieved [5]. Because of this high number of spores and the inevitable consequences, there is a strong demand for non-sporulating commercial strains.

Spontaneous sporeless mutants of basidiomycetes have been found from natural populations in *Coprinopsis cinerea* [6], *Schizophyllum commune* [7], *Lentinula edodes* [8], *Agrocybe salicacola* [9], *Pleurotus ostreatus* [10] and *Pleurotus pulmonarius* [11]. Sporeless mushroom strains have also been generated successfully by mutagenesis using chemical treatment and UV irradiation in *C. cinerea* [12], *Agrocybe cylindracea* [13], *P. pulmonarius*, *P. ostreatus* [14, 15], *Pleurotus eryngii* [16], *Pleurotus florida* and *Pleurotus sajor-caju* [17]. These sporeless mutants are important as breeding material in developing sporeless strains for commercial cultivation. It appeared, however, difficult to restore yield and quality in these mutants to an acceptable level by breeding and there have been, to our knowledge, only three sporeless strains commercially produced: *P. ostreatus* SPOPPO [18, 19], *P. eryngii* [20] and *A. cylindracea* [13]. For the *P. ostreatus* SPOPPO variety, a breeding program based on a spontaneous sporeless *P. ostreatus* mutant, ATCC58937 (F42 x 11; [10]) and commercial variety HK35 was performed. In this breeding program, Baars *et. al.* [21] studied the inheritance of the sporeless phenotype (100% reduction of spores) and concluded that it is recessive, and mapped in both constitutive haploid genomes of the sporeless mutant, on the same chromosome and at the same locus. The sporeless phenotype further mapped on the same chromosome as the *A* mating-type. The breeding program yielded a commercially acceptable sporeless strain although it has a somewhat deviating morphology, in particular, the fruiting bodies show a disturbed orientation. The intention of the present research was to identify and characterize the gene(s) responsible for the sporeless phenotype in *P. ostreatus*. This will facilitate breeding for additional sporeless varieties, allowing a more accurate selection and reduction of linkage drag. If the identified gene(s) similarly participate in sporulation in other (edible) mushroom forming fungi, these could also be used to develop sporeless varieties in other species.

The offspring (spores) in mushrooms are the outcome of meiosis and it has been shown previously that blockage of this process can also eliminate or reduce the production of spores. Deletion of the meiotic genes *DMC1* and *SPO11* impaired sporulation in *P. ostreatus* and *C. cinerea* respectively [22, 23]. In yeast, the *RAD51* gene is involved in meiosis and its *P. ostreatus* homolog showed elevated expression levels in lamellae/basidia, although expression levels are similar when compared between a sporulating strain and the sporeless mutant [24]. Another meiotic gene, *MER3*, is required for synaptonemal complex formation and a null mutant reduces spore formation dramatically in *C. cinerea* [25]. Okuda *et. al.* [26] describe a defective *STPP1* gene, a *P. pulmonarius MSH4* homolog, to be responsible for the absence of spores in a sporeless mutant. Here we show that the absence of spores in the commercial *P. ostreatus* variety SPOPPO is caused by a defect *MSH4* homolog (defined as *poMSH4* for simplicity), that has been disrupted by an insertion of a transposon-like fragment. A cytological study showed that meiosis in the *P. ostreatus* strain SPOPPO is blocked at metaphase I. We also show that an artificial disruption of the *MSH4* homolog of *S. commune* (defined as *scMSH4*), a non-*Pleurotus* sp., diminished sporulation to less than 0.1% of the wild type.

## Materials and methods

### Strains and mapping population

Two *Pleurotus ostreatus* strains were used: a normal sporulating dikaryotic (Sp$^+$dikaryon) strain N001 [27] with its two constituent monokaryons PC9 (Sp$^+$hap1) and PC15 (Sp$^+$hap2) and a non-sporulating dikaryotic mutant ATCC58937 (F42 x 11; [10]) with its two constituent monokaryons EP25 (Sp$^-$hap1) and EP57 (Sp$^-$hap2). Both monokaryons (Sp$^-$hap1 and Sp$^-$hap2)

are carriers of the sporeless trait, resulting in a homozygous locus for this recessive trait. A mapping population was generated by isolating 188 monokaryotic progeny (BKK population) of a cross between Sp⁺hap2 and Sp⁻hap2. Sp⁺hap2 was selected as a parent for the mapping population since its whole genome was sequenced (http://genome.jgi.doe.gov/PleosPC15_2; [28]). This population was used to determine segregation of Single Nucleotide Polymorphism (SNP) markers and Cleaved Amplified Polymorphic Sequences (CAPS) markers and to map the phenotypes sporelessness, disturbed orientation of fruiting bodies and *A* and *B* mating-type. All strains used, were maintained in vials containing perlite in 1% malt extract and 2.5% glycerol and stored in liquid nitrogen. For short term storage, strains were maintained on malt extract agar (MEA; 1% malt extract and 2% agar) in slants at 4˚C. Vegetative mycelium was grown on MEA at 24˚C.

The sequenced monokaryotic *Schizophyllum commune* strain H4-8 (FGSC #9219) and its isogenic derivatives H4-8b, c and d were used as wild-type strains. For generating knock-outs, the dikaryotic Δ*ku80* strain was used (H4-8/H4-8b background; [29]). In this strain the *ku80* gene is deleted by integration of a hygromycin resistance cassette, resulting in abolishment of ectopic integration upon transformation. Strains were grown at 25˚C on minimal medium [30]. When needed minimal medium was supplemented with nourseothricin (8 μg/ml), phleomycin (5 μg/ml) or hygromycin (5 μg/ml).

## Phenotyping

The sporeless phenotype was assessed after crossing each individual of the mapping population with the monokaryon Sp⁻hap1. Spawn of all crosses was prepared by inoculating sterile sorghum seeds (70 g/box) with a 2 x 2 cm agar piece from a fully grown petri dish and incubation at 24˚C for 10–11 days. Bags (Polypropylene 3T bags with BN filter; Unicorn Bags) containing 1 kg of wheat straw substrate were inoculated using 35–40 grams of spawn, sealed and incubated at 24˚C for 20 to 30 days in the dark. When fully colonized, fruiting was induce by changing the environmental conditions to 15˚C, 90% relative humidity, max. 600 ppm $CO_2$ and 12/24 hours of light, while a slit was made on each side of the package. Fruiting bodies were evaluated for their orientation and the gills were microscopically examined for presence of spores on the basidia.

## Sequencing, SNP selection and genotyping

High molecular genomic DNA of Sp⁻hap2 was extracted using the DNeasy Plant Kit (Qiagen, Germany) according to the suppliers protocol and sequenced using Illumina Genome Analyzer II Paired-End Sequencing (ServiceXS, The Netherlands). For genotyping, the sequencing reads were mapped against the reference genome assembly of Sp⁺hap2 and Single Nucleotide Polymorphisms (SNP's) were identified using NextGene software (SoftGenetics State College, PA, USA). SNP's were selected, evenly distributed over the whole genome with 100 kb intervals. For scaffold 3, harboring the sporeless locus, SNP's were selected with about 50 kb intervals.

Genomic DNA was extracted from the individuals of the BKK mapping population using the Wizard® Magnetic 96 DNA Plant System (Promega, USA) and genotyped using the Illumina GoldenGate Assay (ServiceXS, The Netherlands). JoinMap 4 software [31] was used for linkage mapping.

For *de novo* assembly of the whole genome sequence of Sp⁻hap2, high molecular genomic DNA was extracted as described by van Peer *et. al.* [32] but using 10–20 mg of lyophilized mycelium per sample. DNA was dissolved in 110 μl TE containing 1 mg/ml RNAse A and after 1–2 hours of incubation at 37˚C, the entire extraction protocol was repeated. The

extracted DNA was subsequently purified using the Genomic DNA Clean & Concentrator Kit (Zymo Research, USA) according the suppliers protocol. Prior to library construction according to the "Procedure & Checklist—Preparing >30 kb Libraries Using SMRTbell® Express Template Preparation Kit" (Pacific Biosciences, USA), DNA was sheared using the Megaruptor. The library was sequenced on the PacBio Sequel using the Sequel 6.0 chemistry and data were collected with SMRT Link v.6.0.0 (Pacific Biosciences, USA). The subreads were assembled using Canu version 1.7 software [33] resulting in 62 contigs with a total assembly size of 35.0 Mbp.

## Construction of a *poMSH4* expression vector and transformation of *P. ostreatus*

Primers were designed based on the Sp⁺hap2 sequence to amplify genomic regions containing at least one identified gene, including a 1 kb region upstream as a promoter region and 500 bp downstream as a terminator region (S1 Table). Target regions were amplified from the Sp⁺hap2 genome using Phusion DNA Polymerase (Finnzymes, Finland) according the suppliers protocol, purified and ligated with the pGEM®-T Easy Vector System (Promega, USA). The correctness of the sequence of the constructs was confirmed by sequencing.

For protoplast generation, mycelium of the non-sporulating dikaryotic mutant ATCC58937 (Sp⁻dikaryon) was grown in liquid culture [34] and digested according to the method as described by Binninger *et. al.* [35] but using a lysing enzyme solution containing 50 mM maleate buffer (pH 5.5), 0.5 M mannitol and 1 mg/ml *Trichoderma harzianum* cell wall lytic enzymes [36]. Transformation of the protoplasts was carried out as described by [37]. The candidate gene constructs were co-transformed with the carboxin selection marker construct pTM1 [38]; 1 μg pTM1 was used in combination with 5 μg candidate gene construct. After regeneration for 7 days, 24 carboxin resistant colonies were transferred to fresh MEA containing carboxin (2 mg/l) and microscopically screened for the presence of clamp connections. Heterokaryotic transformants were screened for the presence of the constructs via PCR using construct specific primers (S2 Table).

For fructification of the transformants, spawn was prepared as described before. For each transformant, about 300 g of wheat straw substrate was mixed with 15–20 g of spawn and transferred to a box (ECO2box white filter, Duchefa Biochemie b.v.). Enough substrate was added to fill the box tightly. Inoculated substrate was incubated at 24°C for 11–22 days. When fully colonized, boxes were transferred to a climate room (15°C, 90% relative humidity, max. 600 ppm $CO_2$, 12/24 hours of light) and the air filter in the lid was cut to induce fruiting. Gills were microscopically examined at 20x magnification for presence of spores using a Zeiss Axio Scope.A1 microscope with Zeiss AxioCam ERc 5s digital camera. Gill tissue and fruiting bodies were collected and immediately transferred to liquid nitrogen. For each transformant with restored sporulation, a tissue culture and spore print were made.

## Construction of a *scMSH4* knock-out in *S. commune*

A protein sequence homolog of poMSH4 was searched in the *S. commune* H4-8 sequence database (http://genome.jgi-psf.org/Schco3/Schco3.home.html). Protein ID 1186310 (http://genome.jgi-psf.org/cgi-bin/dispGeneModel?db=Schco3&id=1186310) was found to have 63% identity with poMSH4 (E value = 0.0). In order to delete the gene encoding this protein, a pDelcas derivative (pDelMSH4) was made following the protocol described by Ohm *et. al.* [39], containing the *S. commune* phleomycin and nourseothricin resistance cassette. The flanking sequences of *scMSH4* were amplified using the primers Δmsh4ufw and Δmsh4urv for the upstream flank and Δmsh4dfw and Δmsh4drv for the downstream flank (S3 Table) and cloned

on either side of the nourseothricin resistance cassette. The Δ*ku80* strain was transformed as described by de Jong *et. al.* [29] with the modification that, instead of using monokaryotic mycelium to obtain the protoplasts, germinated monokaryotic spores from the Δ*ku80* dikaryon were used. Potential candidates, which grew on nourseothricin but not on phleomycin, were analyzed by PCR using primers msh4ufscf and pdkufscr to test insertion in the upstream region and pdkdfscf and msh4dfscr for insertion in the downstream flank (S3 Table).

For fructification, sorghum grains were inoculated with mycelium and incubated at 30°C in the dark until fully colonized. Colonized grains were transferred to small burlap bags, hung in a beaker above a small amount of water, sealed with parafilm and placed at room temperature in the light. After fructification, fruiting bodies were cut from the bag, transferred to 50 ml Greiner tubes and weighted. To each tube 2 ml of 0.1% Tween80 was added and tubes were placed under vacuum for 40 min. After shaking for 15 min., the spore suspension was transferred to Eppendorf tubes and centrifuged for 5 min. at 13,300 rpm. Spores were resuspended in 100 μl and 50 μl of 0.5x $T_{10}E_{0.1}$ for the wild type and ΔΔ*MSH4* respectively. Spores were counted using a Bürker counting chamber.

### RNA extraction, cloning and sequencing

For total RNA extraction, gill tissue was separated from cap tissue of fruiting bodies of *P. ostreatus* N001 (Sp⁺dikaryon) and ATCC58937 (Sp⁻dikaryon), immediately frozen using liquid nitrogen and stored at -80°C until use. RNA was extracted according to the method described by Sokolovsky *et. al.* [40]. Full length cDNA was generated from total RNA with the SuperScript® III First-Strand Synthesis System for RT-PCR (Invitrogen) according to the manufacturers protocol using Oligo(dT)$_{20}$ primers and random hexamers. The full length *poMSH4* was amplified with primer pair MSH4_For (5'-ATGCAAGCCTCTCGTCCAACAAC-3') and MSH4_Rev (5'-TTACATTACAAAGAGCTTTGCTA-3'), cloned with the pGEM®-T Easy Vector System (Promega, USA) and sequenced by GATC Biotech.

### Cytological analysis

For light microscopy, the Giemsa staining method was used as described by Obatake *et. al.* [16] using 4CF-1G Double aldehyde (Formaldehyde 35% 100 ml/l; Glutardialdehyde 25% 40 ml/l; NaOH 2.7 g/l; $NaH_2PO_4.H_2O$ 11.6 g/l) as a fixative. Following staining, the tissue was dehydrated by passing through a series of graded ethanol baths and embedded into Methacrylate using Technovit® 7100 GMA embedding. Sections of 6–8 μm thickness were mounted on glass slides and again stained with 1:25 Giemsa-phosphate buffer solution (pH 7) for at least 2 hours. After staining, the samples were washed with tap water and dried. A coverslip was mounted after applying Euparal to the sample. The tissue was examined using a Zeiss Axio-Phot light microscope with a Leica DFC340 FX digital camera.

## Results

### Identification of candidate genes involved in sporulation of *P. ostreatus*

We previously showed that the sporeless phenotype was recessive and mapped to the same genomic region in both nuclei of the mutant strain ATCC58937; monokaryons Sp⁻hap1 and Sp⁻hap2 [21]. In order to fine-map the region linked to the sporeless phenotype, SNP markers were generated. For this, the sequencing reads of Sp⁻hap2 were mapped against the whole genome sequence of wild-type monokaryon Sp⁺hap2 as a reference, resulting in a total of 212,832 called variants. For genetic mapping, 384 SNP's were selected of which 98 evenly distributed over scaffold 3, harboring the sporeless locus and the *A* mating-type locus. In

addition, 3 CAPS-markers were selected on this scaffold. A mapping population of 188 mono-karyotic offspring of a cross between Sp⁺hap2 and Sp⁻hap2 was used to generate a genetic map, consisting of 12 linkage groups covering 1050 cM, on average 32.7 kb/cM (S1 Fig; Adapted version of the map previously published by Sivolapova *et.al.* [41]). All 188 individuals were crossed with Sp⁻hap1 and cultured in duplicate to be screened for sporulation and orientation of fruiting bodies. The phenotypes *A* mating-type, *B* mating-type, sporelessness and disturbed orientation of fruiting bodies were added to the map. As expected, the sporeless phenotype and the *A* mating-type were part of the same linkage group; linkage group 3 (138.7 cM; Fig 1). The disturbed orientation of fruiting bodies was tightly linked to the sporeless phenotype. Sporelessness mapped between SNP-marker 3_975676 and 3_1066945, a 87 kb region. Additional fine mapping reduced the relevant region to 78.9 kb. Using the Sp⁺hap2 genome annotation (http://genome.jgi.doe.gov/PleosPC15_2; [28]), 27 predicted genes were identified in this region, of which 3 genes were predicted to play a role in transcription, replication, recombination and DNA repair. These were the main candidates for playing a role in spore production (Table 1) based on the observation of a blocked meiosis (see below).

## Sporulation of *P. ostreatus* is restored by transformation of wild type *poMSH4*

Eleven constructs were made representing in total 23 of the 27 Sp⁺hap2 genes from the same 78.9 kb region that was mapped to the sporeless phenotype in Sp⁻ strains, each construct containing up to 5 genes. Protoplasts of the ATCC58937 (Sp⁻ dikaryon) were transformed with each construct, and 24 confirmed transformants per construct were screened for the presence of clamp connections (dikaryons). Clamp connections were observed in 70–90% of the confirmed transformants. Fruiting bodies were cultivated for 5 dikaryotic transformants per construct (in duplicate) and gills of fruiting bodies were microscopically examined for (lack of) spores. A single construct restored sporulation, containing a unique gene and a second gene that was also present in another construct. A final construct containing the only remaining candidate gene restored sporulation in all Sp⁻ dikaryon transformants (Fig 2). Notably, with restoration of sporulation also the proper orientation of the fruiting bodies was restored. The gene restoring sporulation in the Sp⁻ host was identified as a homolog of MutS homolog 4 (*MSH4*), a meiosis specific gene required for reciprocal recombination and proper segregation of homologous chromosomes during meiosis I. The genomic DNA sequence of the Sp⁺hap2 *poMSH4* (JGI protein ID 1101251; https://mycocosm.jgi.doe.gov/cgi-bin/dispGeneModel?db=PleosPC15_2&id=1101251) was 3,864 bp in length. Alignment with the 2,556 bp total RNA sequence of the sporulating *P. ostreatus* N001 strain (Sp⁺ dikaryon) revealed that the *poMSH4* gene contained 26 exons ranging in size between 8 and 300 bp and encoded a protein of 851 amino acids (S1 File). A BLAST search of the *poMSH4* encoded sequence to Sp⁺hap2 revealed the presence of only one copy of this gene.

## In the sporeless *P. ostreatus* strain *poMSH4* is interrupted by a DNA fragment that can be associated with a Copia-type retrotransposon

The genomic *poMSH4* sequence of Sp⁺hap2 was blasted against the Sp⁻hap2 *de novo* assembly (S2 File), resulting in a match with contig 00000008 (S2 File contig 00000008), on which the Sp⁻hap2 *poMSH4* sequence was found to be interrupted by a 6,792 bp DNA fragment. To study the origin and nature of this integrated DNA fragment (from here on called "insert"), a 75 kb region of contig 00000008 surrounding the interrupted *poMSH4* gene (S3 File) was aligned to the well annotated genome of Sp⁺hap2 (http://genome.jgi.doe.gov/PleosPC15_2; [28]). The corresponding region of the annotated genome of Sp⁺hap2 scaffold 3 (Fig 3A; S1

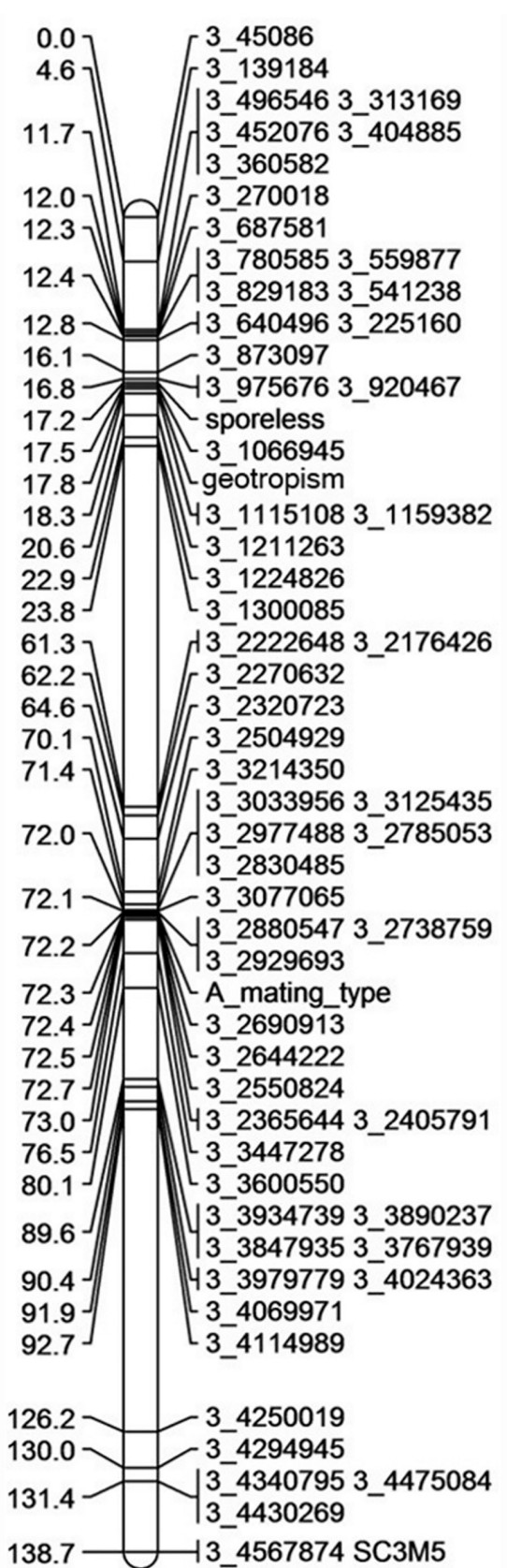

**Fig 1. Linkage group 3 of the genetic linkage map of Sp⁺Hap2 x Sp⁻Hap2.** This linkage group is based on 188 monokaryotic progeny and harbours the phenotypes sporelessness, disturbed orientation of fruiting bodies (geotropism) and the *A* mating-type.

File) showed that *poMSH4* in the sporulating strain was flanked on the left by a gene encoding a YL1 type nuclear protein and on the right by a sequence representing an open reading frame (ORF) encoding a protein of 660 amino acids containing a CxC5/CxC6 like cysteine cluster. Further downstream in the same region on Sp⁺hap2 scaffold 3, a Long Terminal Repeat (LTR)-retrotransposon (LTR-RT) of the Copia type was found together with additional solo-LTRs of the same Copia type RT. The two overlapping ORFs that were present between the LTRs of the Copia type RT contained all known signatures of a Copia type RT (S2A Fig). The ORF encoding a protein of 660 amino acids containing the CxC5/CxC6 like cysteine cluster that was found at the right flank of *poMSH4* in Sp⁺hap2, was missing at the right flank of the interrupted *poMSH4* in Sp⁻hap2. However, the 6,792 bp "insert" in *poMSH4* of Sp⁻hap2 contained an ORF encoding 739 amino acids, that was very similar to the ORF encoding 660 amino acids at the right flank of *poMSH4* in Sp⁺hap2 (Fig 3B). The "insert" further contained three small, slightly overlapping direct repeats of 46 bp and a cluster of microsatellites with a BC₅NWY signature (Fig 3C). An exact but reversed copy of the 6,792 bp "insert" was also

**Table 1. All identified predicted genes located within the 78.9 kb region of the mapped sporeless phenotype.**

| Nr. | Region in PC15 v2.0 (JGI) | KOG Class | Interpro / KOG description |
|---|---|---|---|
| 1 | scaffold_03:1007867–1009396 | Posttranslational modification, protein turnover, chaperones | |
| 2 | scaffold_03:1009681–1010011 | | |
| 3 | scaffold_03:1010157–1011149 | Intracellular trafficking, secretion, and vesicular transport | |
| 4 | scaffold_03:1012134–1013346 | Posttranslational modification, protein turnover, chaperones | Glutathione S-transferase |
| 5 | scaffold_03:1013853–1015734 | General function prediction only | Protein kinase, core |
| 6 | scaffold_03:1016022–1018164 | Carbohydrate transport and metabolism | 6-phosphogluconate dehydrogenase, |
| 7 | scaffold_03:1019029–1020872 | Energy production and conversion | FAD linked oxidase, |
| **8** | **scaffold_03:1022291–1022998** | **Transcription** | **Zinc finger, TFIIS-type** |
| 9 | scaffold_03:1023761–1024511 | | |
| 10 | scaffold_03:1024915–1027652 | Amino acid transport and metabolism | Peptidase M1, membrane alanine aminopeptidase, N-terminal |
| 11 | scaffold_03:1027758–1028630 | Cytoskeleton | WASP-interacting protein VRP1/WIP, contains WH2 domain |
| 12 | scaffold_03:1028626–1029171 | Signal transduction mechanisms | Arf GTPase activating protein |
| 13 | scaffold_03:1029326–1030398 | | |
| 14 | scaffold_03:1030490–1031232 | | |
| 15 | scaffold_03:1031338–1033801 | General function prediction only | DNA-binding protein YL1 and related proteins |
| **16** | **scaffold_03:1033887–1037750** | **Replication, recombination and repair** | |
| 17 | scaffold_03:1038532–1040786 | Intracellular trafficking, secretion, and vesicular transport | |
| 18 | scaffold_03:1047069–1049837 | Posttranslational modification, protein turnover, chaperones | |
| 19 | scaffold_03:1060550–1066783 | Signal transduction mechanisms | |
| **20** | **scaffold_03:1066713–1068943** | **Transcription Regulation** | **Copper fist DNA binding** |
| 21 | scaffold_03:1069152–1073123 | RNA processing and modification | Ketose-bisphosphate aldolase, class-II |
| 22 | scaffold_03:1073942–1076056 | Lipid transport and metabolism | Phosphatidylserine decarboxylase-related |
| 23 | scaffold_03:1076113–1077336 | Posttranslational modification, protein turnover, chaperones | Ubiquitin-conjugating enzyme, E2 |
| 24 | scaffold_03:1077625–1078856 | Coenzyme transport and metabolism | Tetrapyrrole biosynthesis, hydroxymethylbilane synthase |
| 25 | scaffold_03:1079274–1079983 | Posttranslational modification, protein turnover, chaperones | |
| 26 | scaffold_03:1079982–1082435 | Posttranslational modification, protein turnover, chaperones | Glycosyl transferase, group 1 |
| 27 | scaffold_03:1082565–1083606 | Translation, ribosomal structure and biogenesis | |

Positions are based on the Sp⁺hap2 (http://genome.jgi.doe.gov/PleosPC15_2; [28]) reference sequence. Candidate genes indicated in bold.

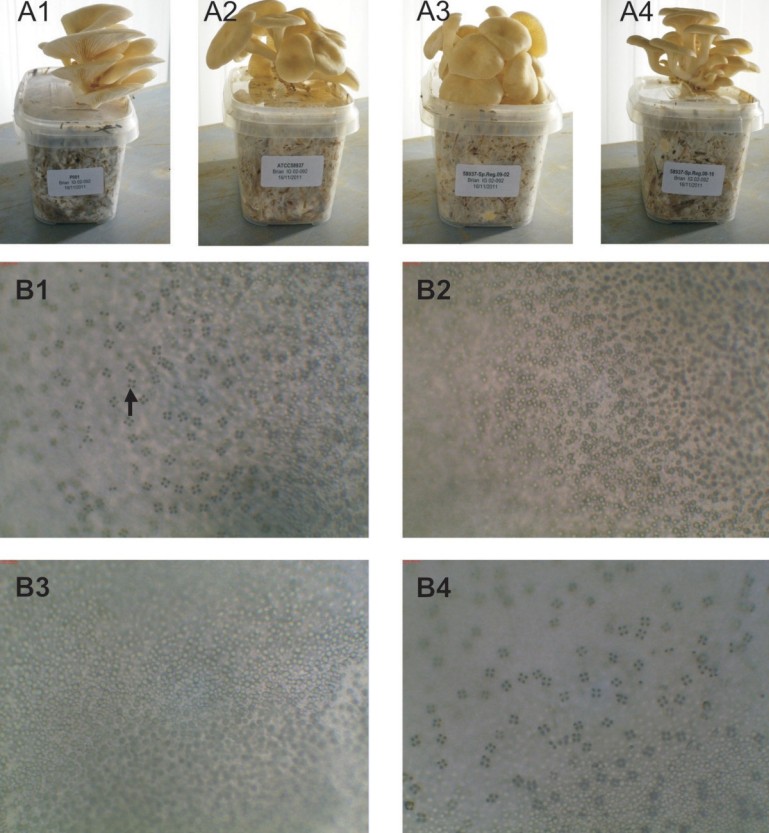

**Fig 2. Fruiting bodies and gill tissue of sporulating and non-sporulating strains.** Fruiting bodies of the sporulating strain N001 (A1), non-sporulating host ATCC58937 (A2), a transformant containing construct Sp. Reg. 09 showing no restored sporulation (A3) and a transformant containing construct Sp. Reg. 08 harbouring the wild type *poMSH4* gene showing restored sporulation (A4). Microscopic pictures of the gill tissue (20x magnification) of the sporulating strain N001 (B1), non-sporulating host ATCC58937 (B2), a transformant containing construct Sp. Reg. 09 (B3) showing no restored sporulation and a transformant containing construct Sp. Reg. 08 showing restored sporulation (B4). The arrow indicates a tetrad; four spores on top of a basidium.

found 23 kb downstream of the interrupted *poMSH4* gene in Sp⁻hap2. Further differences between the Sp⁻hap2 and the Sp⁺hap2 *poMSH4* regions was the presence of multiple copies of the Copia type RT in Sp⁻hap2; 2 reversed intact copies containing all signatures of Copia type RT (S2B Fig) and 3 partial copies surrounding the downstream copy of the "insert" (Fig 3A and 3B). Finally, 2 solo-LTRs of the Copia type RT were found adjacent to the most right, partial Copia type RT. Upstream of *poMSH4* and further downstream of the most right Copia-LTR, the genomes of Sp⁺hap2 and Sp⁻hap2 were identical.

CxC5/CxC6 like cysteine clusters have been reported to be associated with KDZ-type transposons [42]. However, no transposon of the KDZ-type was found within expected distance of the ORFs encoding the CxC5/CxC6 like cysteine cluster. Using the KDZ-type of transposons previously detected in *Coprinopsis cinerea* and *Laccaria bicolor* [42], tBLASTn revealed the presence of 36 KDZ-type transposons, either truncated or seemingly intact, in the annotated genome of Sp +hap2 (http://genome.jgi.doe.gov/PleosPC15_2; [28]). Intact KDZ-type transposons were all preceded by a region encoding a CxC2 cysteine cluster, but never by a CxC5/CxC6 cysteine cluster encoding region (S3 Fig). Regions encoding a CxC5/CxC6 cysteine clusters were also found in high copy numbers throughout the genome. In the wild-type genome (Sp⁺hap2) forty four copies were found, and nearly sixty in the genome of the non-sporulating strain (Sp⁻hap2), either

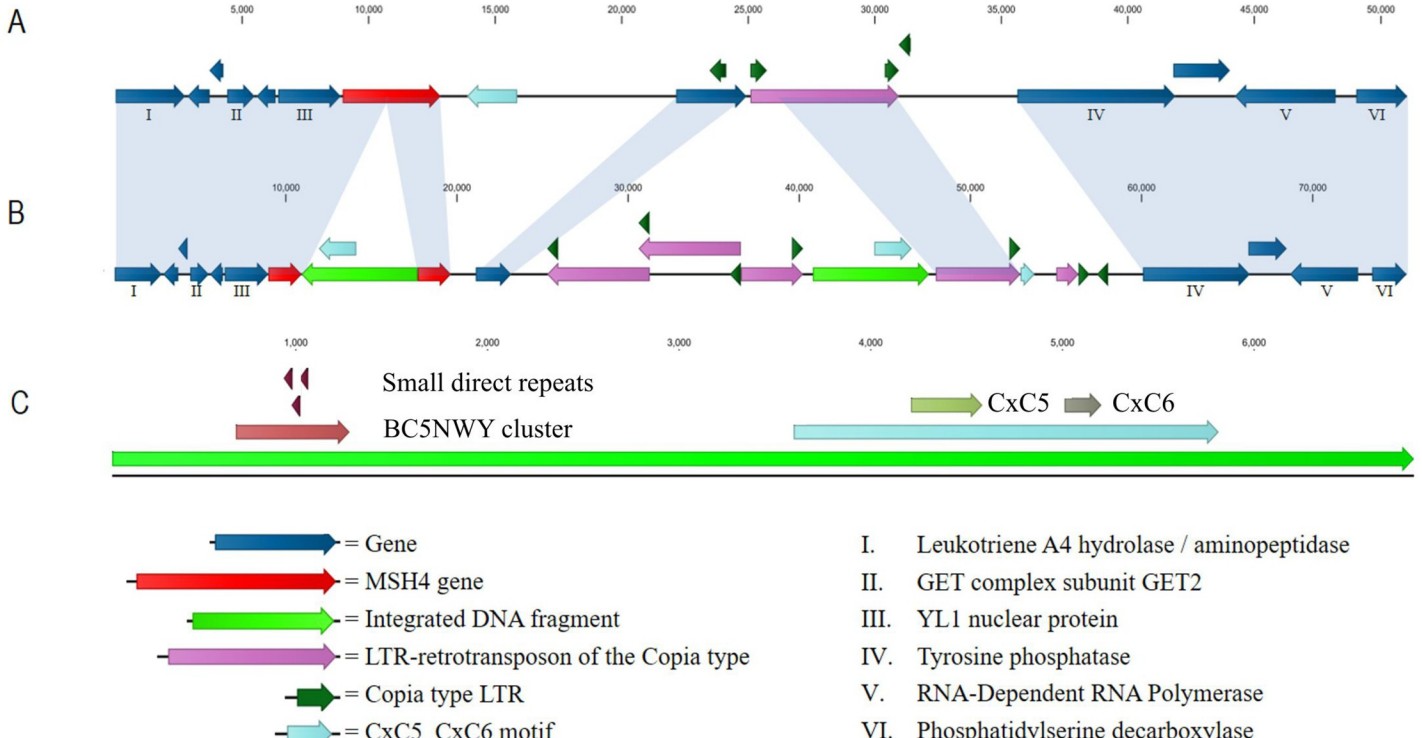

**Fig 3. Representation of the annotated sequence of the *poMSH4* region in Sp⁺hap2 and Sp⁻hap2.** Identical genomic sequences between Sp⁺hap2 (A) and Sp⁻hap2 (B) are indicated with the blue shading. Representation of the 2 identical copies of the integrated DNA fragment of Sp⁻hap2 (C).

complete or truncated. Blasting a Copia-type retrotransposon (RT) consensus sequence and the ORF containing the encoded CxC5/CxC6 cysteine cluster to the sequence of the wild-type and the sporeless strain of *P. ostreatus* showed that CxC5/CxC6 cysteine cluster encoding regions are very often associated with the Copia type RT or the LTRs of this Copia-type RT and vice versa. The Copia-type RT and the regions encoding a CxC5/CxC6 cysteine cluster seem not to be distributed randomly over both genomes but are present, often together, in a limited number of spots.

### Multiple copies of the DNA fragment integrated in *poMSH4* of Sp⁻hap2 are found in the non-sporulating strain as well as the sporulating strain, and are always associated with an intact Copia type retrotransposon

Two additional, full copies of the "insert" were found in the Sp⁻hap2 genome on contig 00003652 (S2 File). Both "inserts" are identical and tail-to-tail oriented, 5,897 bp apart (S4 File). Also in Sp⁺hap2 (http://genome.jgi.doe.gov/PleosPC15_2; [28]) on scaffold 1 (S5 File) and scaffold 7 (S6 File), a full "insert" is found. When aligned, the major difference between these "inserts" and the DNA fragment integrated in *poMSH4* of Sp⁻hap2 is the presence of solo-LTR(s) of the Copia type RT (S4 Fig). But what they all have in common is the presence of at least 1 intact copy of a Copia type RT within a 15 kb distance.

### The sporeless *P. ostreatus* mutant ATCC58937 is blocked in the meiotic metaphase I

To examine cytological characteristics of the non-sporulating *P. ostreatus* mutant ATCC58937 (Sp⁻dikaryon), the presence of all meiotic stages in the basidia was studied in comparison with

the normal sporulating N001 strain (Sp$^+$dikaryon) by light microscopy using HCl-Giemsa stained tissue. In the Sp$^+$dikaryon, all developmental processes in the basidia i.e. karyogamy, stages of meiosis I and II, development of sterigmata and the migration of the daughter nuclei to the basidiospores could be identified (Fig 4A). In the Sp$^-$dikaryon, meiosis seemed to occur normally up to metaphase I. None of the stages of the meiotic division following the metaphase I could be observed (Fig 4B). The sporeless mutant also seemed incapable of sterigmata formation and basidiospore production.

## Construction of a *scMSH4* knock-out in *Schizophyllum commune* results in a strongly reduced sporulation

Protoplasts from germinated spores of a dikaryotic *S. commune* strain Δ*ku80* (H4-8/H4-8b background) were transformed with the *scMSH4* deletion construct pDelMSH4 and selected to be nourseothricin resistant and phleomycin sensitive. Screening for phleomycin sensitivity reduces the number of transformants with the pDelMSH4 plasmid integrated outside the target region. PCR analysis showed that 2 out of 8 transformants contained the deleted *scMSH4* (Δ*MSH4*). Since the transformants were both dikaryotic as a result of unintended protoplast fusion or plasmogamy, they were induced to fructify and monokaryotic progeny was selected on nourseothricin. Monokaryons containing the deleted *scMSH4* and compatible mating-types were selected and crossed to obtain dikaryons homozygous for the interruption. These dikaryons were grown to produce fruiting bodies and their sporulation pattern was studied. In the ΔΔ*MSH4* strain, sporulation was strongly reduced (< 2.6 spores/mg wet tissue) when compared to wild-type dikaryons (1808 spores/mg wet tissue).

To confirm that reduction of sporulation is not related to the *ku80* deleted gene, a mono-karyon of one of the Δ*MSH4* lines was crossed to the isogenic strains H4-8, b, c and d. The crosses that resulted in a dikaryon were selected by looking for clamp connections and were allowed to fructify. Single spore cultures (SSC's) were screened for hygromycin sensitivity (wild type *ku80*) and nourseothricin resistance (Δ*MSH4*) and were crossed among each other. This resulted in dikaryotic lines with the wild type *ku80* gene in combination with a deleted *scMSH4*. The same was done with hygromycin and nourseothricin sensitive colonies resulting in dikaryotic lines with a wild-type *ku80* in combination with a wild-type *scMSH4*. Five dikaryons were chosen from each group and were further investigated. All the dikaryons homozygous for Δ*MSH4* were found to hardly produce any spores when compared to the dikaryons homozygous for wild type *scMSH4* which showed normal sporulation. Except for the strongly reduced sporulation, no phenotypic differences were observed between the dikaryons homozygous for Δ*MSH4* and the dikaryons homozygous for wild type *scMSH4*.

## Discussion

Transformation of candidate genes to the non-sporulating *P. ostreatus* mutant ATCC58937 (Sp$^-$dikaryon) identified the *poMSH4* as the gene responsible for the sporeless phenotype. Comparison of the *poMSH4* sequences of the non-sporulating and the sporulating strain revealed that the *poMSH4* is interrupted by a nearly 7 kb DNA fragment in the non-sporulating mutant. In total 4 copies of this DNA fragment were found in the non-sporulating mutant of which one copy in reversed orientation is located at a 23 kb distance of the copy inserted in *poMSH4*. The other 2 copies are located at a different scaffold, are also tail-to-tail orientated, and are 6 kb apart. Only 2 copies were found in the sporulating strain Sp$^+$hap2, each on a different scaffold. All copies of the "insert" DNA fragment contain a region encoding a CxC5/CxC6 cysteine cluster, previously shown to be associated with KDZ-type transposons [42]. However, as all intact KDZ-type transposons were preceded by a region encoding a CxC2

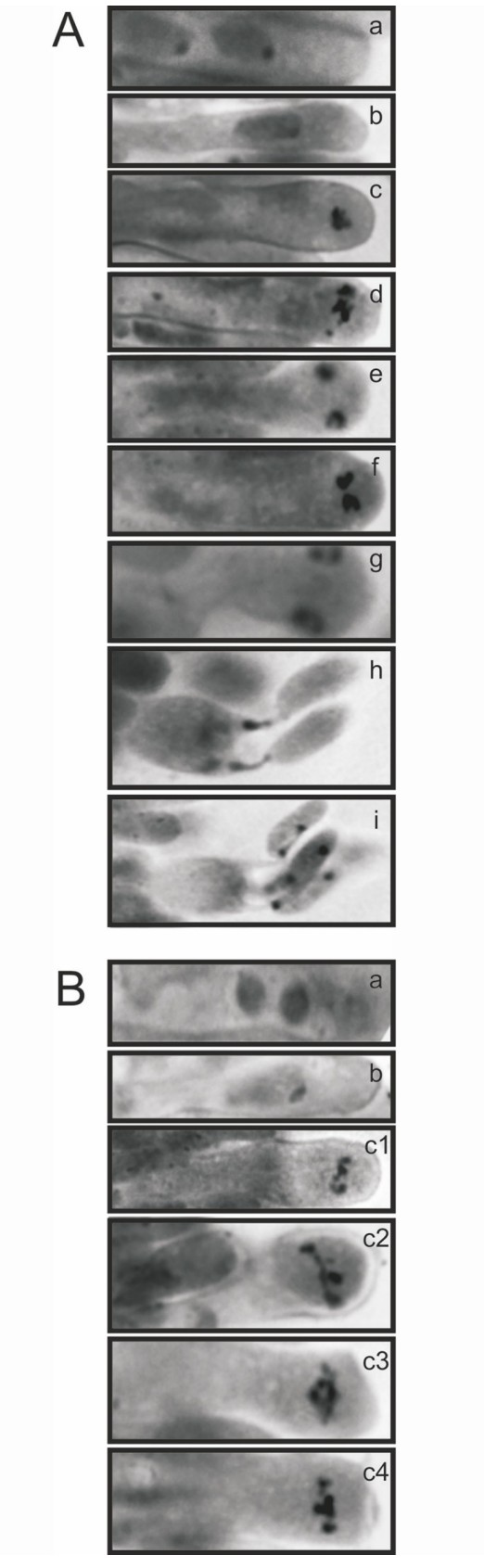

**Fig 4. Microscopic pictures of different stages of meiosis in basidia of the sporulating of sporulating and non-sporulating strains.** Sp⁺ dikaryon (A). Binucleated basidium (a), Nuclear fusion (b), Metaphase I—Anaphase I (c), Metaphase I—Anaphase I (d), Telophase I (e), Meta-anaphase II (f, g), Nuclei migrating to spores (h), Binucleate spores after meiotic division (i), and Sp⁻ dikaryon (B). Binucleated basidium (a), Nuclear fusion (b), Different examples of metaphase I (c1-c4). Beyond this stage no examples have been found indicating that meiosis is halted at this stage (Giemsa staining; 100x magnification).

cysteine cluster, but not by a CxC5/CxC6 cysteine cluster encoding region (S3 Fig), this suggests that in *P. ostreatus* the KDZ-type transposon is not associated with CxC5/CxC6 clusters but possibly with CxC2 clusters. The apparent co-localization of the Copia-type RT and the CxC5/CxC6 cysteine cluster encoding region might indicate a cooperation in transposition between these two sequences. Cysteine cluster rich proteins can be involved in recognition of and binding to specific DNA sequences [43] and might explain the presence in specific regions. Alternatively, transposons might multiply within the region of the first copy. The *poMSH4* containing region of the mutated Sp⁻hap2 has clearly been rearranged compared to the same region in the Sp⁺hap2. It is known that transposable elements can mediate genomic rearrangement in many ways [44]. How the rearrangement in the non-sporulating strain was generated is unclear, but seemingly association of Copia RT type transposons and CxC5/CxC6 cysteine cluster encoding regions, and the presence of such a region encoding a CxC5/CxC6 cysteine cluster in the "insert" that disrupts *poMSH4* do indicate that the Copia type RT might have been involved in the disruption.

Microscopic examination (Fig 4) indicates that meiosis is interrupted late in prophase I or metaphase I, which leads to abolishment of all downstream meiotic events and eventually the absence of spores. This also means that somehow, the outcome of meiosis is controlling the process of spore development. In meiosis, the generation of double strand breaks (DSB) leads to recombination between homologous chromosomes that are resolved into non-crossovers (NCOs) or crossovers (COs). In most organisms the majority of DSB are resolved into NCOs [45]. For COs, two distinct pathways have been found in plants, yeast and animals; Class I and Class II. Class I type COs constitute the majority in most organisms and are characterized by the involvement of the so-called ZMM group of proteins that play a role in recombination and formation of the synaptonemal complex (SC) during meiotic prophase I in budding yeast [46], plants [47] and humans [48]. The MSH4-MSH5 heterodimer is part of this complex, that plays a role in stabilizing single-end strand invasion intermediates that are formed during the early stage of recombination, and that binds to Holliday junctions to facilitate crossover formation [49, 50]. The class II type of COs can also resolve Holliday junctions, but is mediated by Mus81-Eme1 proteins and seems to be independent of the ZMM proteins [51]. MSH4 mutants in budding yeast show a delayed SC formation and full synapsis is achieved only in half of all nuclei, leading to a reduction in spore viability of ca. 50% [52]. Mutation of MSH4 generates a similar phenomenon in *Arabidopsis* [47], where the null mutant shows numerous univalents in meiosis, indicating a strong reduction of chromosome pairing due to a strong reduction in chiasmata. The reduced but not complete absence of progeny following mutations in the ZMM proteins (including MSH4) in these organisms has been explained by the still functioning class II crossover pathway [47]. Mutation of MSH4 in mammals leads to complete sterility [46]. Recently, it has been shown in mice that there is a requirement for an intact MSH4-MSH5 heterodimer in crossing over, and also that MSH4-MSH5 is critical for all crossovers, regardless of their starting route from DSB precursors [53]. The role of MSH4 (and likely MSH5) is thus not exactly the same in all eukaryotes. Here we have observed a mutation in the *MSH4* homolog of *P. ostreatus* that causes a null mutation leading to the complete absence of spores. In *S. commune* (in this article) and in *P. pulmonarius* [26], the *MSH4*

mutation leads to a very strong reduction of spores (both <0.1% of the wild type) and indicates a similar role of the encoded proteins in these organisms, which is strengthened by the presence of similar structural domains in the encoded proteins (S5 Fig). Furthermore it may suggest that in basidiomycetes MSH4 is needed for all type of crossovers or that basidiomycetes do not have a functional (or very inefficient) class II type of CO. The very small number of spores observed in *P. pulmonarius* and *S. commune* might also indicate the existence of additional classes of proteins that can resolve Holliday junctions, but obviously with a very low activity/efficiency.

Most Oyster mushroom varieties show a decrease in yield during prolonged use. The sporeless strain, however, has been continuously cultivated by many growers since its introduction on the market in 2006 and it maintained a high yield. Although speculative, the absence of spores might help explain the stability of the variety. Spores are a known vector for viral diseases in macrofungi [54] including *P. ostreatus* [3]. Different types of viruses have been found in Oyster mushroom crops [3, 55–57] and it is possible that reinfection of crops occur frequently or that even different viral types are accumulated through lingering spores when using different varieties over time. The generation of a sporeless Oyster mushroom variety by introduction of the natural mutation has led to a commercially successful Oyster mushroom variety that is now used by many growers in Europe. The disturbed orientation of Oyster mushrooms within bunches of the sporeless variety could be considered as less beautiful, although most harvested mushrooms are packed as individual mushrooms rendering disturbed orientation irrelevant. Mapping the phenotypes sporelessness and disturbed orientation revealed that both phenotypes are tightly linked. Moreover, introducing a wild type *poMSH4* gene in the sporeless strain restored sporulation as well as the orientation of the fruiting bodies. This indicates that the block in meiosis and absence of spores relate to the disturbed orientation. Gravitropic bending in mushrooms either by stipe or fruiting body is a prerequisite for optimal spore dispersal [58]. How the absence of spores relates to the disturbed orientation remains unknown. In *S. commune* the relationship between sc*MSH4* and fruiting body morphology could not be studied since strain H4-8, used as host for transformation, is already disturbed in its response to gravity. Okuda et al. reported no effect on fruiting body morphology after a knock-out of a *MSH4* homolog in *P. pulmonarius* [26]. However, no bunches of fruiting bodies were shown in their publication. Obatake et al. generated a sporeless *P. eryngii* mutant by UV irradiation [16]. It appeared to be a dominant mutation that also blocked meiosis in metaphase I causing the absence of sterigmata and spores. Interestingly, they observed a deviation from the wild type mushrooms, i.e. mushrooms seemed to have lost a bit of their negative gravitropy and did not grow completely perpendicular in the substrate ("leaning mushrooms" as they call it). This might suggest that, whatever mutation is used to block sporulation, the interruption of meiosis and or the absence of spores can affect the orientation of mushrooms. As for *P. ostreatus*, also for *P. eryngii* the effect is not seen after harvest and packing of individual mushrooms. *MSH4* homologs might thus be a good candidate to generate also sporeless varieties in other edible basidiomycetes. Since the obvious method to generate knockouts, i.e. CRISPR Cas9, is not an breeding method accepted by most consumers, mutants should be obtained in classical ways. Strains of these species containing a mutated *MSH4* homolog may be obtained by screening a natural strain collection for *MSH4* homolog mutants. In addition, the mutant may also be obtained by classical mutagenesis approaches followed by high-throughput screening of this mutant library, as suggested before [26]. This marker can then be used for breeding, following a similar strategy as described for *P. ostreatus* [18, 19, 21] which resulted in a marketable sporeless strain.

## Supporting information

**S1 Fig. Genetic linkage map of Sp⁺Hap2 x Sp⁻Hap2.** The linkage map is based on 188 mono-karyotic progeny of the cross using 387 genetic markers and the phenotypes A mating-type, B mating-type, sporelessness and disturbed orientation of fruiting bodies (geotropism).
(TIF)

**S2 Fig. Identified signatures in the LTR to LTR retrotransposon of the Copia-type.** The retrotransposons located in the *poMSH4* region of Sp⁺hap2 (A) and Sp⁻hap2 (B).
(TIF)

**S3 Fig. Annotation of the ORFs of the KDZ-type transposons found in the annotated sequence of the Sp⁺hap2.** http://genome.jgi.doe.gov/PleosPC15_2.
(TIF)

**S4 Fig. Alignment of all copies of the "insert" found in Sp⁻hap2 and Sp⁺hap2.** Insert_1_1 and insert_1_2 are identical copies, located in the *poMSH4* region of Sp⁻hap2 (contig 00000008) of which insert_1_1 is integrated into *poMSH4*, disrupting the gene. Insert_2_1 and insert_2_2 are identical copies, located on contig 00003652 of Sp⁻hap2. Major difference between the "inserts" located in the *poMSH4* region of Sp⁻hap2 and all the other copies of the "insert" is the absence of solo-LTRs of the Copia type RT (dark green). Next to that, there are some small differences in the number and composition of the small repeat units with the BC5NWY signature (Dark red). The light blue region represents the CxC5/CxC6 cysteine cluster encoding region with the CxC5 domain (light green) and the CxC6 domain (grey).
(TIF)

**S5 Fig. CLUSTAL multiple sequence alignment between the *P. ostreatus poMSH4, P. pulmonarius stpp1* (accession no. AB761293) and *S. commune scMSH4*.** Conserved domains indicated as described by Okuda *et. al.* [26]. The olive green boxes represent the ATP binding site and the stale blue represents the ABC transporter signature motif. Hash tags, upward-pointing arrows, downward-pointing arrows, plus signs and asterisks indicate the Walker A, Walker B, D-loop, Q-loop and H-loop respectively.
(PDF)

**S1 Table. Primer combinations used in vector construction for transformation to *P. ostreatus* strain ATCC58937.** Primers were designed to amplify genes including a 1 kb promoter and 500 bp terminator region using *Pleurotus ostreatus* Sp⁺hap2 as template.
(DOCX)

**S2 Table. Primer combinations for screening *P. ostreatus* transformants for the presence of the candidate gene(s).**
(DOCX)

**S3 Table. Primer combinations for construction of the deletion vector (pDelMSH4) and for screening *S. commune scMSH4* transformants.**
(DOCX)

**S1 File. MSH4_region_PleosPC15_2.** Sequence of the *poMSH4* region of the WT *P. ostreatus* strain (Sp⁺hap2).
(GBK)

**S2 File. Draft assembly of pleosEP57.**
(FASTA)

**S3 File. MSH4_region_EP57.** Sequence of the *poMSH4* region of the *P. ostreatus* mutant strain (Sp⁻hap2).
(GBK)

**S4 File. Insert_2_region_EP57.** Sequence of the region of the *P. ostreatus* mutant strain (Sp⁻hap2) from contig 00003652 containing the additional copies of the "insert".
(GBK)

**S5 File. PC15_insert_Sc01.** Sequence of the "insert" homolog region on scaffold 1 of the WT *P. ostreatus* strain (Sp⁺hap2).
(GBK)

**S6 File. PC15_insert_Sc07.** Sequence of the "insert" homolog region on scaffold 7 of the WT *P. ostreatus* strain (Sp⁺hap2).
(GBK)

## Acknowledgments

The authors thank Patrick Hendrickx for his assistance in generating a genetic linkage map and Ed Hendriks and José Kuenen for their technical assistance in *P. ostreatus* cultivation. Additionally, the authors thank Yoichi Honda and Thomas Mikosch for their expertise and help in setting up a *P. ostreatus* transformation potocol.

## Author Contributions

**Conceptualization:** Anton S. M. Sonnenberg.

**Formal analysis:** Brian Lavrijssen.

**Investigation:** Brian Lavrijssen, Luis G. Lugones, Narges Sedaghat Telgerd.

**Methodology:** Brian Lavrijssen.

**Supervision:** Arend F. van Peer.

**Writing – original draft:** Brian Lavrijssen, Luis G. Lugones, Karin Scholtmeijer, Anton S. M. Sonnenberg, Arend F. van Peer.

**Writing – review & editing:** Johan P. Baars, Karin Scholtmeijer, Anton S. M. Sonnenberg, Arend F. van Peer.

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
