## [Decision Letter · Decision Letter 0]

26 Aug 2020

PONE-D-20-21284

Interruption of an MSH4 homolog blocks meisose in metaphase I and eliminates spore formation in Pleurotus ostreatus

PLOS ONE

Dear Dr. Scholtmeijer,

Thank you for submitting your manuscript to PLOS ONE. After careful consideration, we feel that it has merit but does not fully meet PLOS ONE’s publication criteria as it currently stands. Therefore, we invite you to submit a revised version of the manuscript that addresses the points raised during the review process.

        Based upon the comments of the Reviewers and my own analysis of the manuscript, the following changes must be made in a revision prior to acceptance:

The implicit arguement that the MSH4 homolog is indeed MSH4 by the interchanging use of the terms "MSH4 homolog" and "MSH4" does not appear to be strong. Indeed, the 63% identity between related strains is large considering the relatively high conservation of MSH4. Is there any other information to back this up? In particular address the following three issues: Provide any evidence based on chromosomal synteny of MSH4 between S. commune and P. ostreatus.Provide information on whether the highly conserved structural regions of MSH4 are maintained between the two species?  In this regard, the sequence comparisons between MSH4 of S. commune and P. ostreatus must be directly compared in a Figure together with other known MSH4 genes from a couple of model organisms noting the highly conserved domains. Provide the additional coding details requested by Reviewer 1.Present information regarding whether this MSH4 homolog is the only such homolog in P. ostreatus. Without further evidence, the authors should restrict the nomenclature to “MSH4 homolog” or state tha  “we will define the MSH4 homolog as poMSH4 for simplicity”.2.   Address the fruiting body expression as requested by Reviewer 1.3.   Present the added detail in Figures S1 and S2 requested by Reviewer 1.4.   Discuss the versality of the method more fully as described by Reviewer 1.5.   Make the requested textual changes of Reviewer 1 and the AE (see below) 

   Reviewer 1 was more critical than Reviewer 2 in their evaluation, I concur with the first reviewer and have individually found several additional issues of concern regarding the evidence for the presence of the MSH4 gene in  P. ostreatus (discussed above).

    I have evaluated the manuscript independently. In addition to the issues raised above, I have suggested rewording of parts of the abstract to increase clarity.

Line 36: Change "MSH4" to "the meiotic recombination gene MSH4".Line 38: Change "a MSH4 null mutant" to "the MSH4 null mutant in S. commune" if I understand the meaning properly.Line 39-40: Change ", and when......was observed" with , and the MSH4 null mutant confers an extremely low level of spore formation".Line 42: Change "This confirms MSH4 as a" to "We propose that MSH4 is likely to be.."

Please address all of the issues of Reviewer 1 and the AE responding with the previous and new line number of the changed text. As a note, any statement of a Reviewer's desire to publish the manuscript is inconsistent with the decision making process of PLOS One where the AE makes the decision based upon the factors described in the Editorial Policy.

We look forward to receiving your revised manuscript.

Kind regards,

Arthur J. Lustig, PhD

Academic Editor

PLOS ONE

Journal Requirements:

'The authors received no specific funding for this work'

We note that one or more of the authors are employed by a commercial company: Ceradis B.V.

Reviewers' comments:

Reviewer's Responses to Questions

**Comments to the Author**

1. Is the manuscript technically sound, and do the data support the conclusions?

Reviewer #1: Partly

Reviewer #2: Yes

2. Has the statistical analysis been performed appropriately and rigorously? 

Reviewer #1: Yes

Reviewer #2: Yes

3. Have the authors made all data underlying the findings in their manuscript fully available?

Reviewer #1: Yes

Reviewer #2: Yes

4. Is the manuscript presented in an intelligible fashion and written in standard English?

Reviewer #1: Yes

Reviewer #2: Yes

5. Review Comments to the Author

Reviewer #1: The authors revealed a gene as causal agent of sporeless trait in P. ostreatus cultivar "SPOPPO". In addtion, It is also interesting to clearly show the genetic relationship between sporeless and geotropism trait, which had been mentioned previously. However, this manuscript lacks basic data to their claims, which needs to be fulfilled for publishing.

Major points:

In this paper, the authors concluded that MSH4 is the causative gene of the sporeless trait in the mutants, and that MSH4 has versatility as a breeding target for developing sporeless cultivars in various species including Pleurotus sp. and Agaricales.

1. Please consider adding the following data to ensure the reliability of data of MSH4 in P. ostreatus.

・List the accession number of MSH4 in P. ostreatus.

・Specify the start codon and stop codon, promoter and terminator region in S1 and S2 file. (Actually, I couldn't open the S1 file. Check the format)

・Add the expression analysis by parts of fruiting bodies.

2. This manuscript lacks data to claim its versatility. If the authors intend to claim its versatility in latter revising manuscript, please add the following data and descriptions.

・description of how to use MSH4 as a target in sporeless breeding.

・comparison between sequences (gene and amino acid) of MSH4, derived from P. ostreatus, S. commune and P. pulmonarius.

Minor points:

Line71

Clarify the characteristics of ATCC58937, which is the base of SPOPPO.

・Degree of decrease in sporulation.

・Specify the relationship between sporeless and geotropism, which has been mentioned in the past. Also indicate whether it is positive or negative geotropism.

・About the mutation controlled by a single gene.

Line313

Coprinopsis cinereus→Coprinopsis cinerea

Line457

What does the "cluster" mean? Effect by the geotropism?

Line461 substate→substrate?

Reviewer #2: Interruption of an MSH4 Homolog Blocks Meiosis in Metaphase I and Eliminates Spore Formation in Pleurotus ostreatus PONE-D-20-21284

The manuscript reports MSH4 gene as a breeding target for sporeless strains. The article is weel written and present novelty to the state-of-the-art. The relevance of the advance reported is due to the poor development of breeding programs in cultivated mushrooms compared to other commercial crops such as plants. For instance, it sets the basis for the production of sporeless varieties such as the commonly Pleurotus variety SPOPPO widely cultivated in Europe. Sporeless varieties can prevent allergic sensitivity to airborne conidia, commonly in oyster mushroom in pickers. The genetic techniques employed are accurate and a protocol for Pleurotus transformation is presented. It is suitable for publication in Plos One in the form submitted.

6. PLOS authors have the option to publish the peer review history of their article (what does this mean?). If published, this will include your full peer review and any attached files.

Reviewer #1: **Yes: **Yasuhito Okuda

Reviewer #2: **Yes: **Jaime Carrasco

---

## [Author Response · Author response to Decision Letter 0]

7 Oct 2020

PONE-D-20-21284

Interruption of an MSH4 homolog blocks meiosis in metaphase I and eliminates spore formation in Pleurotus ostreatus

PLOS ONE

Dear Dr Lustig,

Thank you for considering our manuscript for publication in PLOS ONE. The comments of the reviewer 1 and the Academic editor (AE) were addressed point by point below.

• The implicit argument that the MSH4 homolog is indeed MSH4 by the interchanging use of the terms "MSH4 homolog" and "MSH4" does not appear to be strong. Indeed, the 63% identity between related strains is large considering the relatively high conservation of MSH4. Is there any other information to back this up? In particular address the following three issues: 

1. Provide any evidence based on chromosomal synteny of MSH4 between S. commune and P. ostreatus. We apologize for the interchanging use of the terms MSH4 and MSH4 homolog. We meant to discuss the MSH4 homolog without directly claiming it has MSH4 activity. Based on the multiple sequence alignment and the BLAST results (see the following points 2 and 3) we are convinced that we can describe the studied gene as a MSH4 homolog and that additional synteny analysis, which only provides circumstantial evidence, is not needed. 

2. Provide information on whether the highly conserved structural regions of MSH4 are maintained between the two species? In this regard, the sequence comparisons between MSH4 of S. commune and P. ostreatus must be directly compared in a Figure together with other known MSH4 genes from a couple of model organisms noting the highly conserved domains. Provide the additional coding details requested by Reviewer 1. A multiple sequence alignment (MSA) of the poMSH4, the scMSH4 and the P. pulmonarius stpp1 was added. Within this MSA the conserved domains as described by Okuda et. al., 2013 are indicated (S5 Fig). In the manuscript, line 442-444 is added: ‘and indicates a similar role of the encoded proteins in these organisms which is strengthened by the presence of similar structural domains in the encoded proteins (Suppl. Fig S5)’ referring to a supplemental file depicting the multiple sequence alignment. 

3. Present information regarding whether this MSH4 homolog is the only such homolog in P. ostreatus. Without further evidence, the authors should restrict the nomenclature to “MSH4 homolog” or state that “we will define the MSH4 homolog as poMSH4 for simplicity”. A BLAST search of the MSH4 sequence against the P. ostreatus PC15 reference sequence was performed. This resulted in only one hit. To the manuscript the following was added (line 271-272): ‘A BLAST search of the poMSH4 encoded sequence to Sp+hap2 revealed the presence of only one copy of this gene.’ We changed all P. ostreatus MSH4 or MSH4 homolog into “poMSH4” and for the S. commune MSH4 into “scMSH4”. This is indicated in the manuscript in line 38, line 90 and line 93.

• Address the fruiting body expression as requested by Reviewer 1.

Reviewer 1 requests addition of expression analysis (mRNA) of poMSH4 in different parts of the fruiting bodies. We feel that by showing that interruption of poMSH4 results in the sporeless trait and that reintroduction of the DNA encoding poMSH4 in the sporeless strain results in restoration of the phenotype is sufficient evidence that this gene plays a role in spore-formation. Furthermore, the goal of this study was mainly about finding the underlying gene(s) responsible for the sporeless trait. Further studies on the expression, regulation and mechanisms of the encoded protein are outside the scope of the manuscript and is a subject of further studies. 

• Present the added detail in Figures S1 and S2 requested by Reviewer 1.

Reviewer 1 requests specification of the start codon and stop codon, promoter and terminator region in S1 and S2 file. We indicated the start codon, the mRNA and encoded sequence in file S1. As promoter and terminator regions 1 kb upstream and 0.5 kb downstream of the start and stop codon were taken, resp. (indicated in materials and methods). 

Reviewer 1 mentioned that file S1 could not be opened. We do not know what is the problem, we can open it on our computers as a text file. 

• Discuss the versality of the method more fully as described by Reviewer 1.

Reviewer 1 finds that the manuscript lacks data to claim its versatility. If the authors intend to claim its versatility in latter revising manuscript, please add the following data and descriptions:

・description of how to use MSH4 as a target in sporeless breeding. We ended our discussion with “MSH4 homologs might thus be a good candidate to generate also sporeless varieties in other edible basidiomycetes” To clarify and claim the versatility of the method we added the following (lines 476-482): “Since the obvious method to generate knockouts, i.e. CRISPR Cas9, is not an breeding method accepted by most consumers, mutants should be obtained in classical ways. Strains of these species containing a mutated MSH4 homolog may be obtained by screening a natural strain collection for MSH4 homolog mutants. In addition, the mutant may also be obtained by classical mutagenesis approaches followed by high-throughput screening of this mutant library as suggested before [26]. This marker can then be used for breeding, following a similar strategy as described for P. ostreatus [18, 19, 21] which resulted in a marketable sporeless strain. 

・comparison between sequences (gene and amino acid) of MSH4, derived from P. ostreatus, S. commune and P. pulmonarius. Suppl. Fig S5 was added to the manuscript showing the a multiple sequence alignment of the poMSH4, the scMSH4 and the P. pulmonarius stpp1. Conserved domains as described by Okuda et. al. 2013 were indicated. We added line 442-444; “and indicates a similar role of the encoded proteins in these organisms, which is strengthened by the presence of similar structural domains in the encoded proteins (Suppl. Fig S5).”

• Make the requested textual changes of Reviewer 1 and the AE : 

1. Line 38: Change "MSH4" to "the meiotic recombination gene MSH4". Done

2. Line 41: Change "a MSH4 null mutant" to "the MSH4 null mutant in S. commune" if I understand the meaning properly. No this is the null mutant of P. ostreatus. We have clarified this by changing MSH4 to poMSH4 .

3. Line 43-45: Change ", and when......was observed" with , and the MSH4 null mutant confers an extremely low level of spore formation". Done

4. Line 45-46: Change "This confirms MSH4 as a" to "We propose that MSH4 is likely to be.." Done.

• Thank you for stating the following in the Financial Disclosure section:

'The authors received no specific funding for this work'

We note that one or more of the authors are employed by a commercial company: Ceradis B.V. 

This is the current address of the Author Narges Sedaghat Telgerd, the work performed by this author was done when the author was employed by Wageningen University. Therefore, the funding statement does not need to be changed. The current address of this author was removed from the manuscript since this author is not the corresponding author. 

a. Please provide an amended Funding Statement declaring this commercial affiliation, as well as a statement regarding the Role of Funders in your study. If the funding organization did not play a role in the study design, data collection and analysis, decision to publish, or preparation of the manuscript and only provided financial support in the form of authors' salaries and/or research materials, please review your statements relating to the author contributions, and ensure you have specifically and accurately indicated the role(s) that these authors had in your study. You can update author roles in the Author Contributions section of the online submission form. NA

“The funder provided support in the form of salaries for authors [insert relevant initials], but did not have any additional role in the study design, data collection and analysis, decision to publish, or preparation of the manuscript. The specific roles of these authors are articulated in the ‘author contributions’ section.” NA

If your commercial affiliation did play a role in your study, please state and explain this role within your updated Funding Statement. NA

 b. Please also provide an updated Competing Interests Statement declaring this commercial affiliation along with any other relevant declarations relating to employment, consultancy, patents, products in development, or marketed products, etc. NA

Within your Competing Interests Statement, please confirm that this commercial affiliation does not alter your adherence to all PLOS ONE policies on sharing data and materials by including the following statement: "This does not alter our adherence to PLOS ONE policies on sharing data and materials.” (as detailed online in our guide for authors http://journals.plos.org/plosone/s/competing-interests) . If this adherence statement is not accurate and there are restrictions on sharing of data and/or materials, please state these. Please note that we cannot proceed with consideration of your article until this information has been declared. NA

c. Please include both an updated Funding Statement and Competing Interests Statement in your cover letter. We will change the online submission form on your behalf. NA 

• Please amend either the title on the online submission form (via Edit Submission) or the title in the manuscript so that they are identical. Upon resubmitting I will do this (there is a typo in the online submission).

• We note that you have included the phrase “data not shown” in your manuscript. Unfortunately, this does not meet our data sharing requirements. PLOS does not permit references to inaccessible data. We require that authors provide all relevant data within the paper, Supporting Information files, or in an acceptable, public repository. Please add a citation to support this phrase or upload the data that corresponds with these findings to a stable repository (such as Figshare or Dryad) and provide and URLs, DOIs, or accession numbers that may be used to access these data. Or, if the data are not a core part of the research being presented in your study, we ask that you remove the phrase that refers to these data.

The phrase data not shown was removed and references to additional supplemental files S4, S5 and S6 were added (line 341-342). The supporting .gbk files S4, S5 and S6 contain the sequences of the region in Sp-hap2 containing the additional copies of the “insert” (S4), the “insert” homolog regions on scaffold 1 (S5) and scaffold 7 (S6) of Sp+hap2. In addition, we added the draft assembly Pleos_EP57v1.fasta as supplementary file S2. In the manuscript we refer three times to this sequence (line 287, 288 and 340). 

• Further Review Comments to the Author

Reviewer #1: The authors revealed a gene as causal agent of sporeless trait in P. ostreatus cultivar "SPOPPO". In addition, It is also interesting to clearly show the genetic relationship between sporeless and geotropism trait, which had been mentioned previously. However, this manuscript lacks basic data to their claims, which needs to be fulfilled for publishing.

Major points:

In this paper, the authors concluded that MSH4 is the causative gene of the sporeless trait in the mutants, and that MSH4 has versatility as a breeding target for developing sporeless cultivars in various species including Pleurotus sp. and Agaricales.

1. Please consider adding the following data to ensure the reliability of data of MSH4 in P. ostreatus.

・List the accession number of MSH4 in P. ostreatus. The JGI accession number was added (line 267).

・Specify the start codon and stop codon, promoter and terminator region in S1 and S2 file. See answer to previous remarks of AE. 

・Add the expression analysis by parts of fruiting bodies. See answer to previous remarks of AE.

2. This manuscript lacks data to claim its versatility. If the authors intend to claim its versatility in latter revising manuscript, please add the following data and descriptions.

・description of how to use MSH4 as a target in sporeless breeding.

・comparison between sequences (gene and amino acid) of MSH4, derived from P. ostreatus, S. commune and P. pulmonarius. See answer to previous remarks of AE.

Minor points:

Line70-72

Clarify the characteristics of ATCC58937, which is the base of SPOPPO. 

・Degree of decrease in sporulation. The main goal of this study was to find the underlying gene(s) responsible for the sporeless trait of the SPOPPO strain. The strain ATCC58937 was the donor of the sporeless trait (100 % reduction of spores) in SPOPPO. Since sporelessness was mapped at the same location in both parental nuclei of ATCC58937, one of the parental homokaryons was selected to be used in this study. (100 % reduction of spores) was added to the manuscript (line 71-72).

・Specify the relationship between sporeless and geotropism, which has been mentioned in the past. Also indicate whether it is positive or negative geotropism.・About the mutation controlled by a single gene. Disruption of poMSH4 resulted in both the sporeless and disturbed orientation phenotype, reintroduction of the gene results in reversal of both phenotypes. Therefore, the relation we find between both phenotypes is the MSH4 gene. In theory we here deal with negative geotropism that is disturbed, the fruiting bodies normally grow slightly upwards (away from gravity) while the lamellae are oriented downwards. The indication ‘negative’ was added to the text (line 471) .

Line 321 

Coprinopsis cinereus→Coprinopsis cinerea, Changed.

Line 457-458

What does the "cluster" mean? Effect by the geotropism? With “clusters” we mean “bunches”, and we changed the term in the manuscript. 

Line473 substate→substrate? Changed.

Reviewer #2: Interruption of an MSH4 Homolog Blocks Meiosis in Metaphase I and Eliminates Spore Formation in Pleurotus ostreatus PONE-D-20-21284

The manuscript reports MSH4 gene as a breeding target for sporeless strains. The article is well written and present novelty to the state-of-the-art. The relevance of the advance reported is due to the poor development of breeding programs in cultivated mushrooms compared to other commercial crops such as plants. For instance, it sets the basis for the production of sporeless varieties such as the commonly Pleurotus variety SPOPPO widely cultivated in Europe. Sporeless varieties can prevent allergic sensitivity to airborne conidia, commonly in oyster mushroom in pickers. The genetic techniques employed are accurate and a protocol for Pleurotus transformation is presented. It is suitable for publication in Plos One in the form submitted.

Reviewer #1: Yasuhito Okuda

Reviewer #2: Jaime Carrasco

---

## [Decision Letter · Decision Letter 1]

21 Oct 2020

Interruption of an *MSH4* homolog blocks meiosis in metaphase I and eliminates spore formation in *Pleurotus ostreatus*

PONE-D-20-21284R1

Dear Dr. Scholtmeijer,

We’re pleased to inform you that your manuscript has been judged scientifically suitable for publication and will be formally accepted for publication once it meets all outstanding technical requirements.

Kind regards,

Arthur J. Lustig, PhD

Academic Editor

PLOS ONE

Additional Editor Comments (optional):

Reviewers' comments:

Reviewer's Responses to Questions

**Comments to the Author**

1. If the authors have adequately addressed your comments raised in a previous round of review and you feel that this manuscript is now acceptable for publication, you may indicate that here to bypass the “Comments to the Author” section, enter your conflict of interest statement in the “Confidential to Editor” section, and submit your "Accept" recommendation.

Reviewer #1: All comments have been addressed

2. Is the manuscript technically sound, and do the data support the conclusions?

Reviewer #1: Yes

3. Has the statistical analysis been performed appropriately and rigorously? 

Reviewer #1: Yes

4. Have the authors made all data underlying the findings in their manuscript fully available?

Reviewer #1: Yes

5. Is the manuscript presented in an intelligible fashion and written in standard English?

Reviewer #1: Yes

6. Review Comments to the Author

Reviewer #1: Thanks for the detailed revision.

Please be careful to maintain the consistency of the text after the correction.

I look forward to the progress of your research in the future.

7. PLOS authors have the option to publish the peer review history of their article (what does this mean?). If published, this will include your full peer review and any attached files.

Reviewer #1: **Yes: **Yasuhito Okuda

---

## [Editor Report · Acceptance letter]

23 Oct 2020

PONE-D-20-21284R1 

Interruption of an *MSH4* homolog blocks meiosis in metaphase I and eliminates spore formation in *Pleurotus ostreatus*

Dear Dr. Scholtmeijer:

I'm pleased to inform you that your manuscript has been deemed suitable for publication in PLOS ONE. Congratulations! Your manuscript is now with our production department. 

Kind regards, 

on behalf of

Dr. Arthur J. Lustig 

Academic Editor

PLOS ONE